# Current Therapeutic Strategies for Metastatic Triple-Negative Breast Cancer: From Pharmacists’ Perspective

**DOI:** 10.3390/jcm11206021

**Published:** 2022-10-12

**Authors:** Shuanghe Li, Chongyang Bao, Lingli Huang, Ji-Fu Wei

**Affiliations:** 1Department of Pharmacy, Jiangsu Cancer Hospital (Jiangsu Institute of Cancer Research, Nanjing Medical University Affiliated Cancer Hospital), Nanjing 210009, China; 2Department of Clinical Pharmacy, School of Basic Medicine and Clinical Pharmacy, China Pharmaceutical University, Nanjing 211198, China; 3Department of Clinical Pharmacy, School of Pharmacy, Nanjing Medical University, Nanjing 211103, China

**Keywords:** metastatic triple-negative breast cancer, pharmacists’ perspective, chemotherapy, immunotherapy, management strategy

## Abstract

Triple-negative breast cancer (TNBC) is characterized by its high invasiveness, high metastasis and poor prognosis. More than one-third of patients with TNBC will present with recurrence or distant metastasis. Chemotherapy based on anthracyclines and taxanes is the standard treatment strategy for metastatic TNBC (mTNBC). Due to the lack of expression of progesterone receptor, estrogen receptor, and human epidermal growth factor receptor 2, therapies targeting these receptors are ineffective for mTNBC, thus special treatment strategies are required. In recent years, the development of new chemotherapy drugs, targeted drugs and immunotherapy drugs offers good prospects for the treatment of mTNBC. However, as these drugs are still in their infancy, several problems regarding the optimization and management of the clinical application of these new options should be considered. Pharmacists can play an important role in drug selection, drug therapy management, the management of adverse drug reactions and pharmacoeconomic evaluation. In this review, we summarized traditional treatment strategies, and discussed the efficacy and safety of novel agents approved in the last ten years and combination regimens for mTNBC, with the aim of providing management strategies for the clinical management of mTNBC from pharmacists’ perspective.

## 1. Introduction

Triple-negative breast cancer (TNBC) is a particularly aggressive form of breast cancer, characterized by the lack of estrogen receptor (ER), progesterone receptor (PR), and human epidermal growth factor receptor 2 (HER2) [1]. TNBC accounts for approximately 15% to 20% of all breast cancers with a higher recurrence rate, greater metastatic potential, poorer prognosis and lower survival rate than other subtypes of breast cancer [2]. More than one-third of patients with TNBC can present distant metastasis including visceral organs and brain metastases in the third year after diagnosis [3]. As for metastatic TNBC (mTNBC), the median overall survival (OS) is only 12 to 18 months [4]. Chemotherapy has been the only systemic treatment option for mTNBC for many years, and standard chemotherapy regimens are usually based on anthracyclines or taxanes as the first-line treatment.

Chemotherapy drugs can directly kill tumor cells or inhibit the growth and proliferation of tumor cells. However, as well as attacking tumor cells, they also affect normal cells in the body, causing severe systemic toxic side effects. This condition limits the use of chemotherapeutic drugs. Tumor heterogeneity, genetic diversity, and chemoresistance also contribute to the poor response. Due to the lack of ER, PR, and HER2, TNBC is not sensitive to endocrine therapy and HER2 targeted therapy [4]. So, the treatment of mTNBC is limited by the lack of curative therapies, especially for patients who have failed both anthracycline and taxane therapy. With the development of high-throughput sequencing technology, the heterogeneity and complexity of TNBC is better understood [5]. A number of new targets have been developed that may offer bright prospects for the treatment of mTNBC. These novel mechanisms include disruption of the three-dimensional structure of the DNA double strand by trophoblast cell surface antigen 2 (Trop-2) antibodies, inhibition of DNA damage repair, targeting of the phosphoinositide-3 kinase (PI3K) pathway, blocking the binding of tumor cells to T cells, regulating the cell cycle, and so on (Figure 1).

More novel chemotherapy agents, targeted agents, and immunotherapy agents have been shown to be effective for the treatment of mTNBC, such as epothilone analogs, antibody drug conjugates (ADCs), poly (ADP-ribose) polymerase (PARP) inhibitors, programmed cell death 1 ligand 1 (PD-L1)/programmed cell death 1 (PD-1) inhibitors, PI3K inhibitors, and androgen receptor antagonists. The agents approved for the treatment of mTNBC are shown in Figure 2 [6,7]. A detailed analysis of therapeutic trends, promising targets in mTNBC and biomarkers predicting the effectiveness of therapeutic strategies, as well as effective experimental compounds currently in clinical trials have been discussed in detail in other reviews [8,9]. Additionally, Lu et al. summarized the landmark clinical trials in metastatic breast cancer mBC that have shaped the current standard of care of mBC to guide personalized treatment for mBC [10]. However, in clinical practice, several problems regarding optimizing and managing the clinical application of these new options should be considered to improve the safety and efficacy of treatment and quality of patients’ life. The role of pharmacists in the clinical application of new drugs is depicted in Figure 3. In this review, we summarized the traditional treatment strategies, and focused on the clinical management of novel drugs approved for mTNBC in terms of drug efficacy and safety, and their potential combination therapeutic strategies from pharmacists’ perspective, aiming to provide useful information for clinical application. The details of the literature search strategies are described in the Appendix A.

## 2. Traditional Chemotherapy Regimens

Systemic cytotoxic chemotherapy is the cornerstone of therapy in the treatment of patients with mTNBC and the best response to chemotherapy occurs primarily in the first line. Anthracyclines and taxanes-based chemotherapy combination regimens are preferred. Compared with monotherapy, combined chemotherapy regimens, such as the combination of anthracycline and cyclophosphamide and taxane combined with platin, usually have a higher objective remission rate (ORR) and progression-free survival (PFS) for patients with a high disease burden. However, they are more toxic, which adversely impact the patient’s quality of life [11]. Cardiotoxicity is the most serious adverse reaction of anthracyclines, limiting their use. When chemotherapy progresses or patients have poor tolerance to anthracyclines or taxanes, a variety of other cytotoxic drugs are available for the later treatment as monotherapy or combination regimens such as capecitabine, gemcitabine, and cisplatin. It is vital to understand the treatment purpose, prior chemotherapy regimens, disease burden, patient preferences, as well as the posology, in order to choose a suitable treatment strategy for each patient [12].

In recent years, multiple studies showed that platinum-based chemotherapy regimens might play an important role in the chemotherapy of mTNBC as a first- or later-line treatment [13,14], but it seems to cause worse hematologic toxicity [15]. For example, the combination of cisplatin and docetaxel was found to be superior to the capecitabine plus docetaxel regimen as the first-line treatment for mTNBC with improved OS [16], and cisplatin combined with gemcitabine might be more effective than paclitaxel plus gemcitabine and could be an alternative or even preferred first-line chemotherapy strategy for mTNBC patients [17,18]. As the same time, the combination of platinum and PARP inhibitor veliparib also significantly improved PFS and showed a trend towards improved OS for patients with mTNBC [19]. Attention needs to be paid to the possible increased risk of adverse reactions in clinical practice, especially hematological toxicity. In addition, effective molecular markers screening for certain populations likely to benefit from platinum-based chemotherapy regimens may be the means to improve the efficacy, and this represents the focus of current research.

## 3. Novel Chemotherapy Agents

### 3.1. Utidelone

Epothilone analogs have been proved to have potent cytotoxic activity against multidrug resistance cells in preclinical research [20]. They can promote the polymerization of tubulin, stabilize the structure of microtubules and induce cell apoptosis. Ixabepilone is the first epothilone analog approved for the treatment of mBC, either alone or in combination with capecitabine [21]. However, ixabepilone is expensive and discontinuation is common due to its hematological toxicity and liver toxicity [22]. A new drug, utidelone, which showed promising results, has been approved in combination with capecitabine for the second-line and above treatment of mBC in China. Multiple clinical trials have shown that either compared to utidelone or capecitabine alone, the combination of utidelone and capecitabine could significantly improve PFS with 7.9 months versus 5.4 months and 8.44 months versus 4.27 months, respectively. However, peripheral neurotoxicity significantly increased in the combined group, with five times more than that of monotherapy [23,24,25,26] (Table 1). There were no differences in the incidence of neutropenia, leukopenia, or anemia between utidelone plus capeitabine and capeitabine monotherapy [23]. The most common adverse event with utidelone was peripheral sensory neuropathy, which was the main cause of dose reduction or discontinuation.

Based on the efficacy and safety results of utidelone in clinical trials, the following suggestions are made from the perspective of pharmacists. (1) Utidelone can also overcome the resistance to anthracyclines and taxanes with more favorable efficacy and lower toxicity than ixabepilone. Compared to ixabepilone, utidelone is less costly, showing the potential to replace ixabepilone in clinical application in the future. (2) The high incidence of peripheral neurotoxicity of utidelone may be due to the sensitization in patients previously treated with taxanes. The major clinical symptoms of peripheral neuropathy are pain and tingling of the hands and feet, which need to be distinguished from palmar-plantar erythrodysesthesia induced by capecitabine. Clinically, the use of antioxidants such as glutathione, lipoic acid, vitamin C and cell-protective agents, such as amifostine and edaravone, to reduce the accumulation of chemotherapy drugs in the dorsal root ganglion and reduce the oxidative damage to peripheral neurons and nerve fibers may be considered. Moreover, acupuncture, cryotherapy, exercise therapy and ganglioside monosialic acid may also reduce severe peripheral neuropathy, but should be used with caution due to inconclusive efficacy [27,28,29]. (3) The absence of significant myelosuppressive toxicity is the most remarkable feature of utidelone compared to paclitaxel and ixabepilone, which may be a distinct advantage of utidelone. (4) The economics of the drug is an important factor that affects its benefits to patients. Available economic evaluations demonstrated that the cost-effectiveness of utidelone should be taken into account according to the economic level of different regions. Since the use of utidelone is at an early stage, more research is needed to determine whether it is a more economical option that will benefit more patients [30].

**Table 1 jcm-11-06021-t001:** Clinical trials of novel chemotherapeutics approved for mTNBC.

Phase	Intervention	Line of Therapy	Patients	N	Efficacy	Safety (AE ≥ Grade 3)
Phase 2 trial [26]	Utidelone + capecitabine vs. utidelone	Second or later-line	mBC	33	ORR: 42.4% vs. 28.57%; PFS: 7.9 vs. 5.4 months	Peripheral neuropathy: 45.5% vs. 8.6%; hand-foot syndrome: 15.2% vs. 0%; hematologic toxicity: 6.1% vs. 7.1%; myalgia and arthralgia: 15.2% vs. 1.4
Phase 3 trial [23]	Utidelone + capecitabine vs. capecitabine	Second or later-line	mBC refractory to anthracycline and taxane	405	PFS: 8.44 vs. 4.27 months	Peripheral sensory neuropathy: 22% vs. < 1%; palmar-plantar erythrodysaesthesia: 7% vs. 8%
Phase 1/2 trial [31]	Eribulin + olaparib	Second or later-line	Advanced or metastatic TNBC	48	ORR: 37.5%; PFS: 4.2 months; OS: 14.5 months	Leucopenia: 87.5%; anemia: 41.7; neutropenia: 87.5%; febrile neutropenia: 33.3%; diarrhea: 4.2%
Phase 3 trial [32]	Eribulin vs. TPC	Third or later-line	mBC	762	ORR:12% vs. 5%; OS: 13.1 vs. 10.6 months; PFS: 3.7 vs. 2.2 months	Neutropenia: 21% vs. 14%; leucopenia: 12% vs. 5%; fatigue: 8% vs. 10%; peripheral neuropathy: 8% vs. 2%; dyspnea: 4% vs. 2%
Phase 3 trial [33]	Eribulin vs. vinorelbine	Third or later-line	mBC	530	ORR: 30.7% vs. 16.9%; OS: 13.4 vs. 12.5 months; PFS: 2.7 vs. 1.4 months	Total: 88.3% vs. 90.2%; anemia: 2.3% vs. 18.3%; febrile neutropenia: 2.7% vs. 1.2%
Phase 3 trial [34]	Eribulin vs. capecitabine	First-, second-, or third-line	mBC	1102	ORR: 11.0% vs. 11.5%; OS: 15.9 vs. 14.5 months; PFS: 4.1 vs. 4.2 months	Neutropenia: 24.6% vs. 4.2%; leukopenia:13.4% vs. 1.8%; Anemia: 2.0% vs. 0.9%; peripheral neuropathy: 6.4% vs. 0.9%
Phase 2 trial [35]	Eribulin + gemcitabine	First- or second-line	mTNBC	83	ORR: 37.3%; OS: 14.5 months; PFS: 5.1 months	Aminotransferase elevation: 25%; Neutropenia: 23.8%
Phase 2 trial [36]	Eribulin + bevacizumab	Second-line	HER2-negative mBC	58	ORR: 24.6%; OS: 14.8 months; PFS: 6.2 months	Hypertension: 7%; neutropenia: 7%; febrile neutropenia: 7%
Phase 2 trial [37]	Eribulin	First- or second-line	mBC	32	ORR: 43.8%; PFS: 8.3%	Neutropenia: 40.6%; peripheral neuropathy: 12.5%; fatigue: 12.5%; thrombopenia: 6.3%
Phase 2 trial [38]	Eribulin	Second-line	mBC	53	ORR: 20.8%; CBR: 26.4%;	Neutropenia: 35.9%; leukopenia:17%
Phase 2 trial [39]	Camrelizumab + apatinib + eribulin	NR	Advanced TNBC	46	ORR: 37%; PFS: 8.1 months; DCR: 87%	Elevated AST/ALT: 17.4%; neutropenia: 30.4%; leukopenia: 13.0%; thrombocytopenia: 19.6%
Phase 2 trial [40]	Eribulin + gemcitabine vs. Paclitaxel + gemcitabine	First-line	HER2-negative mBC	118	ORR: 48.9% vs. 44.1%; the 6-months PFS rate: 72% vs. 73%	Neutropenia: 57.6% vs. 67.8%; neurotoxicity: 13.6% vs. 45.8%
Phase 1b/2 trial [41]	Eribulin + pembrolizumab	First-line and later-line	mTNBC	167	PD-L1^+^ vs. PD-L1^−^: (1) first-line: ORR: 34.5% vs. 16.1%; (2) second or later-line: ORR: 24.4% vs. 18.2%	Neutropenia: 26%; immune-related AE: 12%

AE: adverse event; TNBC: triple-negative breast cancer; mBC: metastatic breast cancer; TPC: treatment of physician’s choice; ORR: overall response rate; OS: overall survival; PFS: progression-free survival; DCR: disease control rate; CBR: clinical benefit rate.

### 3.2. Eribulin Mesylate

Eribulin has a different mode of action compared to taxanes and vinca alkaloids, as it inhibits dynamic instability by binding to a few high-affinity sites at the growth terminal of microtubules [33]. Its novel and unique mechanism of action, which exerts its antitumor effects mainly by preventing the growth of microtubules without affecting shortening and by segregating microtubulin as non-functional aggregates. Eribulin has strong antitumor activity against a variety of tumor cells and the ability to overcome taxane resistance conferred by β-tubulin mutations [42]. Unlike paclitaxel, it can improve the OS of taxane-resistant relapsed or mBC due to its unique mechanism. The EMBRACE clinical trial showed that eribulin achieved significantly superior OS [32], and was well tolerated in previously treated mBC [33]. It has been approved for the third-line treatment of mBC [43]. Substantial evidences supported eribulin as the third- or later-line treatment of mTNBC [33,44], but there is still controversy over whether eribulin can be used as a first- or second-line treatment for mTNBC. When used as first- or second-line therapy, eribulin monotherapy showed sufficient efficacy and the clinical benefit and tumor control rates were 56.3% and 78.1%, respectively [37]. However, eribulin is comparable to capecitabine monotherapy in OS or PFS for mBC patients [34]. Eribulin in combination with gemcitabine also demonstrated promising antitumor activity as first- or second-line therapy for mTNBC, with an ORR and PFS of 37.3% and 14.5 months, respectively. Moreover, the study also found that compared with BRCA pathogenic variants, BRCA wild-type patients had higher efficacy with an ORR of 41.5% versus 26.7% [35]. Compared with paclitaxel plus gemcitabine, eribulin plus gemcitabine chemotherapy had similar clinical benefits in terms of PFS, but had lower neurotoxicity [40]. In addition, the combination of eribulin and pembrolizumab showed good antitumor effect for both PD-L1 positive and negative tumors in mTNBC, but the ORR of PD-L1 positive tumor patients was higher than that of PD-L1 negative [41]. Neutropenia was the most common grade 3 adverse event. Peripheral neuropathy was the most common adverse event, leading to the discontinuation from eribulin. The details are shown in Table 1.

Although eribulin has been widely used in Western countries for a long time, it has only been approved in China and other Asian countries in recent years. Its use in clinical practice is limited. Based on the efficacy and safety of eribulin, the following points are available from pharmacists’ perspective. (1) Eribulin has similar efficacy to capecitabine, but the incidence of neutropenia, peripheral neurotoxicity and other adverse reactions is higher. Eribulin has no obvious advantage. (2) The combination of eribulin and gemcitabine has a relative advantage compared to paclitaxel plus gemcitabine, especially for BRCA wild-type patients. This may be a new first-line option for mTNBC. (3) Eribulin may be a promising combination partner for immunotherapy as first-line treatment of mTNBC. Future studies could assess the efficacy based on PD-L1 status to describe the certain population that would benefit the most from the combination of therapy. (4) The major enzyme of eribulin metabolism is cytochrome P450 (CYP) 3A4. As a competitive inhibitor, eribulin might affect the CYP3A4 metabolism of several therapeutic agents and lead to increased toxicity, such as paclitaxel, midazolam, and terfenadine. A clinical study of drug-drug interactions showed that eribulin could be safely co-administered with ketoconazole at a 50% reduction in dose [45]. However, there still lacks sufficient in vivo data. (5) Eribulin-related myelosuppression is reversible, not cumulative, which can be managed by dose delay or reduction, or by the use of granulocyte colony-stimulating factor (G-CSF). G-CSF may be considered for primary prevention in patients who have experienced significant hematologic toxicity in previous chemotherapy regimens [46], particularly in patients with hepatic impairment or severely impaired renal function or elderly patients [47]. (6) The incidence of peripheral neuropathy after treatment with eribulin might be relatively low compared to taxanes and vinorelbine. When eribulin was combined with gemcitabine, peripheral neurotoxicity was significantly lower than that of paclitaxel combined with gemcitabine. Eribulin is likely to replace paclitaxel as a drug combined with gemcitabine to reduce adverse reactions.

## 4. Novel Targeted Therapeutic Agents

### 4.1. ADCs

ADCs are novel and highly effective targeted drugs that combine monoclonal antibodies to one or several small cytotoxic drug molecules through linkers. There are three ADCs including sacituzumab govitecan, ladiratuzumab vedotin and trastuzumab deruxtecan, that show obvious clinical efficacy against TNBC. Sacituzumab govitecan was approved as the first ADC by Food and Drug Administration (FDA) and National Medical Products Administration for the treatment of adult patients with mTNBC who have received at least two previous therapies [48]. Trastuzumab deruxtecan was approved by FDA for the treatment of adult patients with unresectable or metastatic HER2-low (immunohistochemistry (IHC) 1^+^ or IHC 2^+^/in situ hybridization (ISH)-) breast cancer who have received prior chemotherapy in the metastatic setting or developed disease recurrence during or within six months of completing adjuvant chemotherapy [49].

Sacituzumab govitecan is an ADC combining a monoclonal humanized antibody targeting Trop-2 as a high ratio of the DNA topoisomerase I inhibitor SN-38 [50]. Sacituzumab govitecan can bind to Trop-2 on the surface of tumor cells and enter tumor cells through target-mediated endocytosis [51]. This mechanism may reduce toxic effects in normal tissues that do not express the target, and can both minimize off-target toxicity and maximize the effect of the drug on tumor cells expressing Trop-2. Sacituzumab govitecan showed good efficacy in TNBC patients in both Phase I/II and Phase III clinical trials as a monotherapy with significant improvement of PFS, OS, and ORR (35.5% in sacituzumab govitecan group versus 5% in chemotherapy group), especially in patients with high or medium Trop-2 expression [50,52,53,54]. The most common adverse events of any grade were neutropenia, diarrhea, nausea, alopecia, fatigue, and anemia. Myelosuppression and diarrhea were more frequent grade 3 adverse events in patients with sacituzumab govitecan than with chemotherapy [53]. Moreover, neutropenia was associated with uridine diphosphate glucuronosyltransferase family 1 member A1 (UGT1A1)*28 homozygosity and the incidence increased with the increase in the copy number [52] (Table 2).

Unlike sacituzumab govitecanis, trastuzumab deruxtecan is a HER2-targeted ADC with a topoisomerase I inhibitor payload, which was initially proved to be effective against HER2-positive breast cancer [64]. However, since a phase 1 study showed promising clinical antitumor activity and a controlled safety profile in HER2-negative breast cancer patients [55], more clinical studies have found significant efficacy of trastuzumab deruxtecan in HER2-low (IHC 1^+^ vs. IHC 2^+^/ISH^−^) mBC with significantly longer PFS and OS than the physician’s choice of chemotherapy (10.1 months versus 5.4 months, 23.9 months versus 17.5 months, respectively) [56]. The most common drug-related adverse events of trastuzumab deruxtecan were nausea, fatigue, thrombocytopenia, decreased appetite, and alopecia; all of these occurred at higher rates than the chemotherapy group and might be positively related to the dosage [55].

From pharmacists’ perspective, there are several points regarding the use of sacituzumab govitecan. (1) Sacituzumab govitecan and trastuzumab deruxtecan can be used as third-line or later treatment for mTNBC and is the preferred treatment for refractory TNBC. However, further studies on the exact efficacy of trastuzumab deruxtecan in mTNBC patients are critical. (2) There are no biomarkers that can clearly predict the efficacy of sacituzumab govitecan. The expression level of Trop-2 may be a biomarker associated with the efficacy of sacituzumab govitecan and high expression of Trop-2 may predict the benefit of this drug [65]. Moreover, gBRCA1/2m status was initially found to be a potential predictive biomarker of response to sacituzumab govitecan in breast cancer [54]. Nectin-4, which is only expressed in TNBC, may also be a new promising prognostic biomarker and target. Efficiency was found to be dependent on both the dose and the nectin-4 tumor expression level in the anti-nectin 4 ADC and (N41mab-vcMMAE) was developed [66]. (3) SN-38 is the cytotoxic component of sacituzumab govitecan, which is a highly potent topoisomerase I inhibitor and metabolite of irinotecan. Some SN-38 is easily converted to lower active forms, such as SN-38 glucuronide (SN-38G), a product of glucuronic acid, which can cause diarrhea by spreading bacterial enzymes through the enterohepatic circulation. Low levels of SN-38G in serum and SN-38/SN-38G in the gut are expected to reduce the risk of severe diarrhea in patients [67]. The concomitant use of drugs that affect the activity of UGT1A1 such as irinotecan should be avoided, because it is involved in the metabolism of SN-38 to SN-38G and may increase the frequency and severity of adverse reactions [68]. (4) When patients first develop grade 4 neutropenia for a minimum 7-day duration or grade 3 febrile neutropenia, a 25% reduction in sacituzumab govitecan dose and administration of G-CSF is recommended in the prescribing information released by the FDA [69]. However, the evidence for prophylactic use of G-CSF is very limited and preliminary prophylactic use is not recommended [70]. (5) Evidence suggests an increased risk of adverse reactions in patients with the UGT1A1*28 variant genotype. The appropriate dose for these patients is unknown. Therefore, it is necessary to explore whether there is a minimum threshold for dose reduction in patients with UGT1A1*28 purex, thereby reducing toxicity [70].

### 4.2. PARP Inhibitor

PARP is a new target in cancer therapy, which can repair single-strand DNA damage via the base excision pathway. PARP inhibitors can treat cancers with defective homologous recombination DNA repair defects such as BRCA1/2 mutation [71]. Approximately 20% of TNBC patients have a germline defect in BRCA1/2 [72]. Olaparib and talazoparib have been proved to significantly improve the clinical response rate compared to chemotherapy in patients with unselected TNBC with germline BRCA-mutated (gBRCAm) [57,58,59,62]. However, patients without BRCA mutations had a low response rate in unselected TNBC. Moreover, olaparib was found to be ineffective in patients with wild-type BRCA1/2 mTNBC [60]. Currently, olaparib and talazoparib have been approved by the FDA and European Medicines Agency (EMA) for the treatment of patients with gBRCAm HER2-negative mBC, and veliparib is still in clinical trials [61,73,74]. Talazoparib has higher activity and selectivity than other PARP inhibitors [63], and shows more effectiveness and less toxicity as second-line treatment than capecitabine, gemcitabine and vinorelbine [62,75]. Olaparib and talazoparib may cause severe hematologic toxicities such as myelodysplastic syndrome and acute myeloid leukemia, which mostly occur within the first three months. However, there are differences in the rates of adverse events between the two drugs, especially higher rates of nausea and vomiting in patients treated with olaparib and higher rates of alopecia and anemia in patients treated with talazoparib [59,62]. Nausea and vomiting have been clearly demonstrated to be the most common adverse effects of PARP inhibitors. The National Comprehensive Cancer Network concludes that olaparib carries a moderate to high risk of vomiting. Moreover, olaparib has significantly stronger rare and severe hematological toxicity than other PARP inhibitors, including myelodysplastic syndrome, bone marrow failure and acute myeloid leukemia, warranting caution in clinical drug therapy [76]. The details are shown in Table 2.

PARP inhibitors resistance has been proved to be a thorny problem [77]. Prior acquired resistance to platinum agents may promote PARP inhibitors resistance. Currently, the main way to overcome resistance to PARP inhibitors is in combination with other drugs, such as immunotherapy drugs and chemotherapy drugs [78]. Olaparib plus carboplatin or durvalumab have been proved to be a safe and effective option by increasing OS in TNBC [78,79]. Moreover, HSP90 inhibitor onalespib has also been proved to be a potential partner of olaparib, and exhibited antitumor activity against BRCA1-mutated patient-derived xenograft models with acquired PARP inhibitors’ resistance and the models with retinoblastoma-pathway alterations in preclinical and phase 1 studies [80]. Therefore, finding the best combination of PARP inhibitors and reducing toxicity is a problem to be solved.

Based on these findings, the following points are of concern from pharmacists’ view. (1) In mTNBC patients with gBRCAm and PD-L1 negative, PARP inhibitors may be first-line treatment options recommended by the European Society for Medical Oncology (ESMO) and American Society of Clinical Oncology (ASCO) guidelines [6,7]. However, there is insufficient evidence to determine the optimal sequencing of PARP inhibitors with other regimens. This should be considered based on disease burden, prior treatment response, drug toxicity, and PD-L1 status. (2) gBRCAm may be an effective biomarker of PARP inhibitors, and genetic testing for pathogenic variants in BRCA1/2 is recommended for all TNBC patients, regardless of age, family history or BC subtype [6]. (3) PARP inhibitors are easy to use orally and show good compliance for most patients. Nonetheless, it may interact with a variety of drugs due to CYP450 enzyme inhibition or induction, such as phenytoin, carbamazepine, and ciprofloxacin, which may reduce PARP inhibitor efficacy or be associated with serious adverse drug reactions [81]. Prescribing information issued by the FDA recommends that olaparib should be avoided in combination with CYP3A inhibitors and inducers. If this cannot be avoided, the dose should be reduced [82]. Evaluation of the patient for all concomitant medications and guidance on specific foods and medications that the patient should avoid is an important measure prior to the use of PARP inhibitors. (4) In addition, for oral drugs, some patients may use drugs incorrectly, especially those with poor compliance. They should be instructed to use drugs correctly, including usage and dosage, and advised of common adverse reactions and treatment measures, and replacement methods for missed drugs. (5) It is also important to note that the order of administration of combination therapy may affect the efficacy. There is evidence that when olaparib is combined with carboplatin, pre-exposure of carboplatin can lead to intracellular accumulation of olaparib and reduce bioavailability [79], so olaparib should be administered first. (6) Due to the high risk of vomiting, 5-hydroxytryptamine 3 receptor antagonists such as ondansetron, dolasetron and granisetron should be used to prevent nausea and vomiting [62]. (7) In the first three months of treatment, patients’ complete blood counts should be closely monitored. If a patient with hematologic toxicity does not recover within 28 days or continues to lose cells after dose adjustment, further investigation must be performed, and whether to discontinue the PARP inhibitor must be considered [83].

## 5. Immune Checkpoint Inhibitors (ICIs)

TNBC is more likely to benefit from ICIs therapy than other breast cancer subtypes [84] due to a large number of tumor-infiltrating lymphocytes (TIL), high level of PD-L1 expression and greater number of nonsynonymous mutations, which can generate tumor-specific neoantigens and activate specific T cells, so as to produce antitumor immune responses [85]. Optimal responses to immunotherapeutic drugs depend on appropriate priming of T cells through antigen processing and presentation and co-stimulation of activated dendritic cells. A PD-L1-positive tumor, first-line immunotherapy, high TIL level, non-liver metastasis, and high CD8^+^ T-cell infiltrating level were found to predict high ORR in ICIs treatment [86]. Compared with standard chemotherapy, ICIs therapy demonstrated more favorable responses in the first-line setting [87,88,89,90,91]. We have summarized the data from the clinical trials of ICIs treatment for mTNBC in Table 3.

Atezolizumab has been shown to be active and safe in a small group of patients with TNBC. Atezolizumab combined with nab-paclitaxel could significantly improve the median PFS and OS in untreated mTNBC patients, especially in PD-L1 positive patients [94,99,100]. This combination regimen has been approved for advanced and metastatic TNBC patients with positive PD-L1 (combined positive score, CPS ≥ 10) by EMA [101]. However, atezolizumab combined with paclitaxel has not shown improved PFS and OS in mTNBC patients, indicating that paclitaxel should not replace nab-paclitaxel when atezolizumab is combined with chemotherapy [95]. Pembrolizumab monotherapy showed good antitumor activity and efficacy may be correlated with the treatment line, with the ORR for first-line therapy significantly higher than second- or later-lines therapy (21.4%, 18% and 5.7%, respectively) [87,91,93]. Compared to pembrolizumab monotherapy, pembrolizumab plus chemotherapy showed more significant improvement in PFS among patients with mTNBC with a CPS of 10 or more, and provided consistent benefit irrespective of the chemotherapy partner in subgroups. With PD-L1 negative or CPS < 1, there was no significant clinical benefit compared with placebo plus chemotherapy [92]. Pembrolizumab plus chemotherapy has also been approved by the FDA for the treatment of locally recurrent or mTNBC patients with PD-L1 positive [102]. Nab-paclitaxel might enhance antitumor activity and has promising activity when combined with pembrolizumab as the first-line treatment of mTNBC patients [103]. In addition, the combination of pembrolizumab and eribulin was generally well tolerated and had greater antitumor activity than either the drug alone for patients with mTNBC [41]. Immune-related adverse events (irAEs) caused by inhibition of PD-1/PD-L1 inhibitors are common both in clinical trials and clinical applications with varied reactions among individuals, including immune-related skin adverse reactions, gastrointestinal adverse events, hepatitis, pneumonia, and thyroid dysfunction [104]. Nevertheless, the majority of toxicities are low in severity. Postponing and stopping ICIs treatment can alleviate the adverse reactions [104].

The following points regarding ICIs treatment of mTNBC are of interest to pharmacists. (1) The combination of pembrolizumab/atezolizumab and nab-paclitaxel may be a first-line treatment strategy for mTNBC patients with PD-L1 positive tumors. (2) Pembrolizumab plus eribulin may be a promising option with good efficacy and manageable safety for patients with mTNBC. (3) In patients with mTNBC, a comprehensive assessment of biomarkers that can predict response to ICIs is critical. In detail, PD-L1 expression levels may be positively correlated with efficacy, but it is not sufficient to assess this alone [105]. Several biomarkers may also predict the efficacy of ICIs, such as PD-L1 expression, the number of TIL, number of CD4^+^ T cells, tumor mutational burden (TMB) status, microsatellite instability (MSI) status, mismatch-repair deficiency (dMMR), age, etc. [106]. Tumor cells with high TMB can produce more tumor-associated antigens that can be recognized by T lymphocytes to activate an immune attack. Anti-PD-1 treatment can make T cell response more effective, so tumors with high TMB may be more sensitive to anti-PD-1 treatment. In addition, solid tumors in MSI-H/dMMR are usually immunogenic and have extensive T-cell infiltration, which makes them highly responsive to ICIs. CD8^+^ T cells are effector cells that directly kill tumor cells in tumor microenvironment, but are inhibited by the PD-1/PD-L1 pathway [107]. Therefore, the number of TIL can be used as a predictor to assess the immune responses. (4) Paclitaxel may not be substituted for the nab-paclitaxel as a combination of atezolizumab for the treatment of mTNBC. The reasons for the different results may be the difference in the clinical activity between paclitaxel and nab-paclitaxel, the heterogeneity of TNBC, and previous treatment [108]. Compared to paclitaxel, nab-paclitaxel, which uses albumin nanoparticles as a carrier is more targeted to tumors. At the same time, nab-paclitaxel can be directly dissolved in normal saline, avoiding the severe allergic side effects caused by traditional paclitaxel injection containing a large amount of surfactant [109], and there is no need to use glucocorticoid or antihistamine drugs for anti-allergic pretreatment of patients in clinical practice. Therefore, it should be considered that the concomitant use of paclitaxel with steroids may reduce immunosuppressant efficacy. (5) Corticosteroid therapy is usually the first-line treatment against irAEs, as recommended by the ASCO guideline. Grade 3 toxicities generally warrant suspension of ICIs and the initiation of high-dose corticosteroids [104]. Discontinuation of ICIs is recommended at the onset of grade 4 toxicity, except for endocrine disorders that have been controlled by hormone replacement [104]. However, due to the immunosuppressive effect and more adverse reactions such as infections, the use of corticosteroids needs careful management. The initial treatment with 1 mg/kg daily methylprednisolone could provide similar efficacy with reduced risk of steroid-related complications when compared with higher-dose regimens [110]. Therefore, low-dose corticosteroids, close follow-up, and regular inspection are important measures to reduce irAEs [111]. (6) Early identification and continuous monitoring of irAEs is critical. Exploring biomarkers to predict and early identify irAEs risk is an urgent issue. Clinically, circulating blood cell counts are a biomarker that is accessible in current clinical practice. Other biomarkers such as cytokines, autoantibodies, serum proteins, intestinal microbiota, microRNA and gene profiling need more research before they can be adopted in practice [112]. (7) Due to the non-specific characteristics of irAEs symptoms, for many patients and medical staff, they are difficult to detect and recognize. Pharmacists can provide consulting services and rigorous pre-education to patients and medical staff can promote the understanding, identification and management of irAEs, as well as reduce the possibility of discontinuation of treatment due to irAEs.

## 6. Conclusions

Chemotherapy has always been the backbone therapy for mTNBC. The development of new chemotherapy drugs, targeted drugs and immunotherapy drugs provides more options for the treatment of mTNBC. Utidelone and eribulin provide an effective treatment option for mTNBC patients who are resistant to chemotherapy such as anthracyclines and taxanes. Targeted therapeutics ADCs and PARP inhibitors have the advantages of high selectivity, high destruction and controllable toxicity. In particular, ICIs will occupy a place in the first-line treatment of mTNBC. The study of optimal combined therapy strategies will be the focus of the treatment of mTNBC in the future. As the same time, due to the heterogeneity of mTNBC, the challenges are to undertake studies on more selected patient populations and to identify reliable biomarkers to predict treatment response. Pharmacists can play an important role because these drugs are still in their infancy in the treatment of TNBC. They can ensure the effectiveness, safety, economy and suitability of drugs in clinical application through drug therapy management, pharmacoeconomic evaluation, drug application evaluation and toxicity management research.

## Figures and Tables

**Figure 1 jcm-11-06021-f001:**
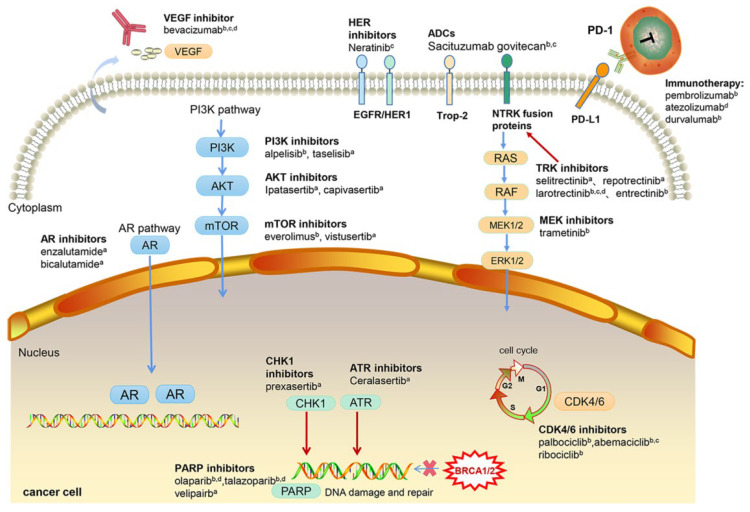
The mechanism of novel therapeutic drugs in mTNBC clinical studies. ADCs: antibody–drug conjugates; AR: androgen receptor; PARP: poly (ADP-ribose) polymerase; Trop-2: trophoblast cell-surface antigen 2; PD-1: programmed cell death-1; PD-L1: programmed cell death-ligand 1; VEGF: vascular endothelial growth factor; CHK: checkpoint kinase; CDK: cyclin-dependent kinase; ATR: ataxia telangiectasia and Rad3-related kinase; PI3K: phosphatidylinositol 3-kinase; MEK: mitogen-activated extracellular signal-regulated kinase; EGFR: epidermal growth factor; HER: human epidermal growth factor receptor. (a) In clinical trial; (b) Registered by Food and Drug Administration; (c) Registered by National Medical Products Administration; (d) Registered by European Medicines Agency.

**Figure 2 jcm-11-06021-f002:**
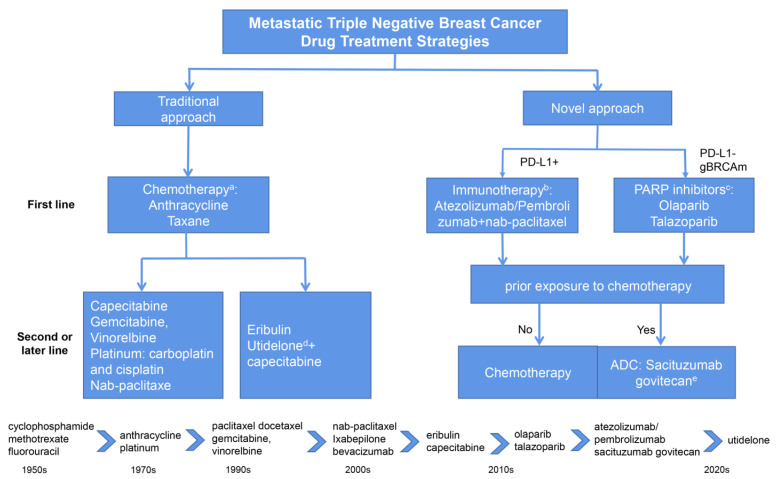
The brief history of the development of drugs and current therapeutic strategies approved for mTNBC. mTNBC: metastatic triple-negative breast cancer; PARP: poly (ADP-ribose) polymerase; PD-L1: programmed death-ligand 1. (a) Single-agent chemotherapy is the preferred option, but combination regimens may be offered for symptomatic or immediately life-threatening disease (recommended by the European Society for Medical Oncology and American Society of Clinical Oncology). (b) Atezolizumab plus nab-paclitaxel was approved by the European Medicines Agency; pembrolizumab plus nab-paclitaxel and other chemotherapy was approved by the Food and Drug Administration. (c) There is insufficient evidence to determine the optimal sequencing of PARP inhibitors with other treatments. (d) Utidelone was approved by the National Medical Products Administration. Utidelone plus capecitabine was recommended by the Chinese Society of Clinical Oncology. (e) Food and Drug Administration and National Medical Products Administration approved.

**Figure 3 jcm-11-06021-f003:**
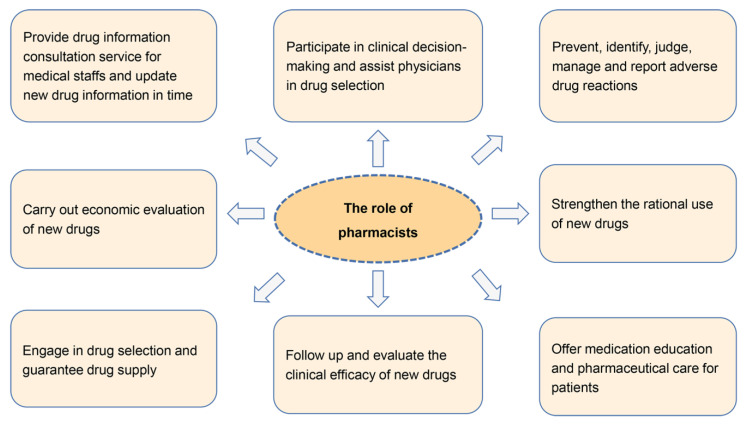
The role of pharmacists in the clinical application of novel antitumor drugs.

**Table 2 jcm-11-06021-t002:** Clinical trials of novel targeted therapeutic agents approved for mTNBC.

Phase	Intervention	Line of Therapy	Patients	N	Efficacy	Safety (AE ≥ Grade 3)
Phase 3 trial [53]	Sacituzumab govitecan vs. single-agent chemotherapy	Second- or later-line	mTNBC	468	ORR: 35% vs. 5%; OS: 12.1 vs. 6.7 months; PFS: 5.6 vs. 1.7 months	Neutropenia: 51% vs. 33%; leukopenia: 10% vs. 5%; anemia: 8% vs. 5%; febrile neutropenia: 6% vs. 2%; diarrhea: 10% vs. <1%
Phase 1/2 trial [52]	Sacituzumab govitecan	Third- or later-line	mTNBC	108	ORR: 33.3%; OS: 13.0 months; PFS: 5.5 months	Neutropenia: 39%; leukopenia: 16%; diarrhea: 13%; vomiting and hypophosphatemia: 10%;
Phase 1 trial [55]	Trastuzumab deruxtecan	NR	advanced/metastatic HER2-low–expressing breast cancer	54	ORR: 37.0%; median duration of Response: 10.4 months	Total: 63.0%; decreases in neutrophil; platelet; WBC counts; anemia; hypokalemia; AST increase; decreased appetite; diarrhea
Phase 3 trial [56]	Trastuzumab deruxtecan vs. the physician’s choice of chemotherapy	Second- or later-line	HER2-low metastatic breast cancer	557	PFS: 10.1 months vs. 5.4 months; OS: 23.9 months vs. 17.5 months	Total: 52.6% vs. 67.4%; Neutropenia: 13.7% vs. 40.7%; anemia: 8.1% vs. 4.7%; Nausea: 4.6% vs. 0%; fatigue: 7.5% vs. 4.7%
Phase 3 trial [57,58]	Olaparib vs. single-agent chemotherapy	Third-line	HER2-negative mBC with gBRCAm	302	ORR: 59.9% vs. 28.8%; OS: 19.3 vs. 17.1 months; PFS: 7.0 vs. 4.2 months	Anemia: 36.6% vs. 50.5%
Phase 2 trial [59]	Olaparib	First-line	TNBC	32	ORR: 56.3% ORR: (BRCA mutations vs. not BRCA mutations): 88.9% vs. 11.1%	Fatigue: 3%
Phase 2 trial [60]	Olaparib	NR	Advanced ovarian carcinoma or TNBC	91	NR	Nausea, fatigue, vomiting, decreased appetite
Phase 3 trial [61]	Veliparib + carboplatin + paclitaxel vs. carboplatin + paclitaxe	Third- or later-line	Advanced HER2-negative BC with gBRCAm	513	ORR: 75.8% vs. 74.1%; OS: 33.5 vs. 28.2 months; PFS: 14.5 vs. 12.6 months	Neutropenia: 81% vs. 84%; anemia: 42% vs. 40%; thrombocytopenia: 40% vs. 28%
Phase 3 trial [62]	Talazoparib vs. single-agent chemotherapy	Second- or later-line	Advanced BC with gBRCAm	431	ORR: 62.6% vs. 27.2%; PFS: 8.6 vs. 5.6 months	Primary anemia: 55% vs. 38%; nonhematologic: 32% vs. 38%
Phase 2 trial [63]	Talazoparib	Third-line	Advanced BC with gBRCA m	84	ORR (TNBC): 26%	Anemia; thrombocytopenia; neutropenia

AE: adverse event; mTNBC: metastatic triple-negative breast cancer; ORR: overall response rate; OS: overall survival; PFS: progression-free survival; gBRCAm: germline BRCA-mutated; PARP: polyadenosine diphosphate-ribose polymerase; BC: breast cancer; HER2: human epidermal growth factor receptor 2; NR: not reported.

**Table 3 jcm-11-06021-t003:** Clinical trials of ICIs approved for mTNBC.

Phase	Intervention	Line of Therapy	Patients	N	Efficacy	Safety (AE ≥ Grade 3)
Phase 3 trial [91]	Pembrolizumab vs. chemotherapy	Second- or third-line	mTNBC	622	(1) CPS ≥ 10: ORR: 18% vs. 9%; OS: 12.7 vs. 11.6 months (2) CPS ≥1: ORR: 12% vs. 9%; OS: 10.7 vs. 10.2 months	Anemia: 1% vs. 3%; decreased white blood cells: <1% vs. 5%; decreased neutrophil count: <1% vs. 10%; neutropenia: 0% vs. 13%;
Phase 3 trial [92]	Pembrolizumab + chemotherapy vs. chemotherapy	First-line	mTNBC	847	(1) CPS ≥ 10: PFS: 9.7 vs. 5.6 months; (2) CPS ≥ 1: PFS:7.6 vs. 5.6 months	Total: 68% vs. 67% immune-mediated AE: 5% vs. 0%
Phase 2 trial [87]	Pembrolizumab	Second- or later-line (Cohort A)	mTNBC	170	(1) Total: ORR: 5.3%; DCR: 7.6%; OS: 9.0 months; (2) PD-L1^+^: ORR: 5.7%; DCR: 9.5%	Diarrhea 1.8%; increased alanine aminotransferase: 1.2%; immune-mediated AE: type 1 diabetes mellitus; pneumonitis;
Phase 2 trial [93]	Pembrolizumab	First-line (Cohort B)	mTNBC	84	ORR: 21.4%; DCR: 23.8%; OS: 18.0 months; PFS: 2.1 months	Total: 9.5%; immune-mediated AE: rash
Phase 3 trial [94]	Atezolizumab + nab-paclitaxel vs. nab-paclitaxel	First-line	mTNBC	902	(1) Total: OS: 21.0 vs. 18.7 months; (2) PD-L1^+^: OS: 25.0 vs. 18.0 months	Neutropenia: 8% vs. 8%; peripheral neuropathy: 6% vs. 3%; decreased neutrophil count: 5% vs. 4%; fatigue: 4% vs. 3%
Phase 3 trial [95]	Atezolizumab + paclitaxel vs. paclitaxel	First-line	PD-L1-positve mTNBC	651	ORR: 63% vs. 55%; OS: 22.1 vs. 28.3 months; PFS: 6.0 vs. 5.7 months	Total: 53% vs. 46%
Phase 2 trial [96]	Pembrolizumab + radiotherapy	NR	mTNBC	17	ORR: 17.6%; PFS: 2.6 months; OS: 8.25 months	Fatigue; lymphopenia; infection
Phase 2 trial [97]	Niraparib + pembrolizumab	NR	mTNBC	55	ORR: 21%; PD-L1^+^: 32%; PFS: 2.3 months	Anemia: 18%; thrombocytopenia: 15%; fatigue: 7%
Phase 2 trial [98]	Pembrolizumab plus enobosarm	NR	AR-positive mTNBC	16	ORR: 13%; OS: 25.5 months; PFS: 2.6 months	Musculoskeletal ache: 6%; dry skin: 6%; diarrhea: 6%

AE: adverse event; HR: hazard ratio; mTNBC: metastatic triple-negative breast cancer; ORR: overall response rate; OS: overall survival; PFS: progression-free survival; AR: androgen receptor; PD-L1: programmed cell death 1 ligand 1; DCR: disease control rate; CPS: combined positive score; NR: not reported.

## Data Availability

Not applicable.

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
