# Peer review of "Current Therapeutic Strategies for Metastatic Triple-Negative Breast Cancer: From Pharmacists’ Perspective"

_jcm, 2022, doi:10.3390/jcm11206021_

Round 1
Reviewer 1 Report
In this manuscript, Li et al. reviewed the literature regarding the Current therapeutic strategies of metastatic triple‑negative breast cancer. In recent years, the development of new chemotherapy drugs, targeted drugs and immunotherapy drugs offers good prospects for the treatment of metastatic triple‑negative breast cancer. They summarized traditional treatment strategies, and discuss the efficacy and safety of novel agents approved in recent ten years and combination regimens for metastatic triple‑negative breast cancer, as well as management strategies from pharmacists’ perspective. This review is important for metastatic triple‑negative breast cancer treatment. However, it is still required for explaining the following issues:
1. A subset of patients with triple‑negative breast cancer is HER2-low expression breast cancer. Trastuzumab deruxtecan showed efficacy in patients with HER2-low metastatic breast cancer. Therefore, trastuzumab deruxtecan should be added in the review.
2. The authors should provide the detailed literature searching strategy in the supplementary information.
3. The adverse effects and predictive biomarkers for sacituzumab govitecan should be discussed.
4. Assessment of other potential predictive biomarkers for immune checkpoint therapy in metastatic triple‑negative breast cancer would be very useful (TMB status, MSI status, TILs, etc.).
5. The authors should compare the difference of the outcomes between this review and a similar review [PMID: 30825473, 34364700, and 34065837] in the discussion section.
6. Figure 1-3: The definition is not high enough. Figures with high dpi should be provided.
7. Although the manuscript is understandable, there are still some grammatical and spelling errors throughout which could be easily corrected.
Reviewer 2 Report
This review paper summarized conventional chemotherapeutic strategies of mTNBC, and other novel agents were also introduced with regards to the treatment efficacy and safety. Although this summary is not complete, the authors tried to deal with comprehensive information of a variety of mechanisms of anti-cancer agents. The message addressing the summarized role of pharmacists was clear. However, I recommend the authors revise some grammatical errors throughout the whole manuscript (including the abstract).
Author Response
Dear Reviewer,
Thank you very much for giving us this opportunity to revise the manuscript (Manuscript ID: jcm-1856351). We have carefully read your comments and revised the manuscript accordingly. The changes were marked in deep red font in the manuscript (Please see the attachment).
Yours sincerely,
Lingli Huang, MPharm.
Jiangsu Cancer Hospital, Jiangsu Institute of Cancer Research, Nanjing Medical University Affiliated Cancer Hospital
E-mail: lingli_huang@njmu.edu.cn.
Ji-Fu Wei, PhD.
Jiangsu Cancer Hospital, Jiangsu Institute of Cancer Research, Nanjing Medical University Affiliated Cancer Hospital; Nanjing Medical University
E-mail: weijifu@njmu.edu.cn.
Responses to the reviewer 2’ comments
This review paper summarized conventional chemotherapeutic strategies of mTNBC, and other novel agents were also introduced with regards to the treatment efficacy and safety. Although this summary is not complete, the authors tried to deal with comprehensive information of a variety of mechanisms of anti-cancer agents. The message addressing the summarized role of pharmacists was clear. However, I recommend the authors revise some grammatical errors throughout the whole manuscript (including the abstract).
Response: Thanks for your comment. We have corrected grammatical errors and polished the language of the article.
Reviewer 3 Report
The paper outlines current strategies in the treatment of TNBC, which is not very novel, but this 'pharmacist' perspective has increased my interest as a practitioner- oncologist. Overall, the article is well written, logic, and transparent. Tables and figures help explain different mechanisms and are well designed. I do not have any major criticism, my comments are just suggestions to make this paper more valuable.
1. Language is understandable and clear. I am just not sure what the authors mean in the sentence at line 187 'Elibrine has no obvious advantage'; Is there a typo?
2. Major comments:
- At some point there should be a reference to the guidelines eg.ESMO/ASCO/NCCN with a short description of what is the current standard of care. For oncologists who do not treat TNBC this general information is crucial.
- In Figure 2 in the first line treatment there are several chemotherapeutic agents. In the main text, there is nothing about the use of cyclophosphamide, 5-FU and methotrexate. The figures should correspond to the main text. To my knowledge and following the ESMO recommendation (Gennari et al. 2021, Annals of Oncology) the standard 1st line are anthracyclines and taxanes plus combination in case of rapid organ failure. In my opinion, cyclophosphamide, 5-FU, and methotrexate should be removed or placed in subsequent lines. Moreover, there is a lack of reference for these strategies. The part with ‘novel approach’ should be simplified and the first decision point are molecular alterations, followed by adequate agents. The horizontal line between PD-L1+ and BRCAm is not understandable. Please divide the whole figure precisely into 1st line agents and 2nd line agents.
- Tables 1, 2 and 3 are well designed and contain a lot of information. For practitioners, the number of clinical trials (NCT) is not crucial information, as the authors give a reference for each report, but the phase of clinical trials is important. Please, put in the first column phase of the clinical trial in every intervention rather than NCT.
- This ‘pharmacits’ perspective’ should be the greatest strength of the article and something what I was looking for during the reading. I like the idea that at the end of each paragraph the authors put in points these impressions. However, most of this information is purely clinical rather than pharmacological. There is a lack of drug interactions, pharmacokinetics, pharmacodynamics, and administration in special circumstances (liver/kidney failure)- and this is what clinicians mostly expect from pharmacists and pharmacologists. Such reports would correspond to the title and would make this paper more useful. I suggest putting some additional information in a systematic order in every paragraph, such as: 1) adverse events & coping with them, 2) important drug interactions, 3) pharmacokinetics/pharmacodynamics, 4) administration in special circumstances (liver/kidney failure), 5) cost-effectiveness/pharmacoeconomics. The authors may, of course, modify and adjust these points according to their own ideas.
3. Minor comments
- In Figure 1 please add symbols indicating which agents are currently registered by EMA/FDA or other;
- Please add references to Figure 2;
- In Tables 1,2, and 3 please specify who approved these treatment modalities- FDA?EMA?
- Please specify the mechanism of action of erilbulin. In line 157 there is only written that it acts differently than taxanes and vinca alkaloids;
- Please add information on differences in adverse events between the described PARP inhibitors;
- in line 263 please specify where olaparib and talazoparib have been approved (reference for EMA/FDA or other);
- In lines 347-352 the reference for immune-related adverse events management is missing. The reference to any guidelines would be better than to KEYNOTE-355 trial;
4. Conclusion- The paper is interesting and well-written. In my opinion, it deserves publication in the Journal of Clinical Medicine. Some of the above-mentioned improvements may increase its novelty and make it more useful for practitioners.
Author Response
Dear Reviewer,
Thank you very much for giving us this opportunity to revise the manuscript (Manuscript ID: jcm-1856351). We have carefully read your comments and revised the manuscript accordingly. The changes were marked in blue font in the manuscript (Please see the attachment).
Yours sincerely,
Lingli Huang, MPharm.
Jiangsu Cancer Hospital, Jiangsu Institute of Cancer Research, Nanjing Medical University Affiliated Cancer Hospital
E-mail: lingli_huang@njmu.edu.cn.
Ji-Fu Wei, PhD.
Jiangsu Cancer Hospital, Jiangsu Institute of Cancer Research, Nanjing Medical University Affiliated Cancer Hospital; Nanjing Medical University
E-mail: weijifu@njmu.edu.cn.
Responses to the reviewer 3’ comments
- Languageis understandable and clear. I am just not sure what the authors mean in the sentence at line 187 'Elibrine has no obvious advantage'; Is there a typo?
Response: Thanks for your comment. We are sorry that there is a spelling mistake and we have corrected it as "Eribulin has no obvious advantage" in line 207, page 7.
Major comments:
- At some point there should be a reference to the guidelines eg.ESMO/ASCO/NCCN with a short description of what is the current standard of care. For oncologists who do not treat TNBC this general information is crucial.
Response: Thanks for your meaningful suggestion. We have added the guidelines and described the current standard of care in lines 341-348, 352-354 page 10, and 408-411, 441-445, page 13.
- In Figure 2 in the first line treatment there are several chemotherapeutic agents. In the main text, there is nothing about the use of cyclophosphamide, 5-FU and methotrexate. The figures should correspond to the main text. To my knowledge and following the ESMO recommendation (Gennari et al. 2021, Annals of Oncology) the standard 1st line are anthracyclines and taxanes plus combination in case of rapid organ failure. In my opinion, cyclophosphamide, 5-FU, and methotrexate should be removed or placed in subsequent lines. Moreover, there is a lack of reference for these strategies. The part with ‘novel approach’ should be simplified and the first decision point are molecular alterations, followed by adequate agents. The horizontal line between PD-L1+ and BRCAm is not understandable. Please divide the whole figure precisely into 1st line agents and 2nd line agents.
Response: Thanks for your valuable suggestion. We have carefully revised the figure and carefully added figure notes and references in Figure 2.
- Tables 1, 2 and 3 are well designed and contain a lot of information. For practitioners, the number of clinical trials (NCT) is not crucial information, as the authors give a reference for each report, but the phase of clinical trials is important. Please, put in the first column phase of the clinical trial in every intervention rather than NC
Response: Thank you for your meaningful suggestion. We have removed NCT and added the phase of clinical trials in the first column in Tables 1, 2 and 3.
- This ‘pharmacits’ perspective’ should be the greatest strength of the article and something what I was looking for during the reading. I like the idea that at the end of each paragraph the authors put in points these impressions. However, most of this information is purely clinical rather than pharmacological. There is a lack of drug interactions, pharmacokinetics, pharmacodynamics, and administration in special circumstances (liver/kidney failure)- and this is what clinicians mostly expect from pharmacists and pharmacologists. Such reports would correspond to the title and would make this paper more useful. I suggest putting some additional information in a systematic order in every paragraph, such as: 1) adverse events & coping with them, 2) important drug interactions, 3) pharmacokinetics/pharmacodynamics, 4) administration in special circumstances (liver/kidney failure), 5) cost-effectiveness/pharmacoeconomics. The authors may, of course, modify and adjust these points according to their own ideas.
Response: Thanks for your valuable suggestion. We have added drug interactions, pharmacokinetics, pharmacodynamics and administration in special circumstances, and then adjusted information in a systematic order in (lines 202-227, page 7; lines 273-301, page 9; lines 339-369, page 10-11; lines 414-460, page 13).
Minor comments
- In Figure 1 please add symbols indicating which agents are currently registered by EMA/FDA or other;
Response: Thanks for your comment. We have added symbols of Figure 1.
- Please add references to Figure 2;
Response: Thanks for your comment. We have added references to Figure 2. References are as follows.
[6] Gennari, A.; Andre, F.; Barrios, C.H.; Cortes, J.; de Azambuja, E.; DeMichele, A.; Dent, R.; Fenlon, D.; Gligorov, J.; Hurvitz, S.A.; et al. ESMO Clinical Practice Guideline for the diagnosis, staging and treatment of patients with metastatic breast cancer. Ann Oncol 2021, 32, 1475-1495.
[7] Moy, B.; Rumble, R.B.; Come, S.E.; Davidson, N.E.; Di Leo, A.; Gralow, J.R.; Hortobagyi, G.N.; Yee, D.; Smith, I.E.; Chavez-MacGregor, M.; et al. Chemotherapy and Targeted Therapy for Patients With Human Epidermal Growth Factor Receptor 2-Negative Metastatic Breast Cancer That is Either Endocrine-Pretreated or Hormone Receptor-Negative: ASCO Guideline Update. J Clin Oncol 2021, 39, 3938-3958.
- In Tables 1,2, and 3 please specify who approved these treatment modalities- FDA? EMA?
Response: Thanks for your comment. We took into account that many treatment regimens in these clinical trials were not approved. Therefore, we have added the approval department for these drugs in the text (line 139, page 5; line 234, 237, page 7; line 312, page 10; line 392, 402, page 12).
- Please specify the mechanism of action of erilbulin. In line 157 there is only written that it acts differently than taxanes and vinca alkaloids;
Response: Thanks for your comment. We have added the mechanism of action of erilbulin in lines 177-179, page 6.
Its novel and unique mechanism of action exerts its antitumor effects mainly by preventing the growth of microtubules without affecting shortening and by segregating microtubulin as non-functional aggregates.
- Please add information on differences in adverse events between the described PARP inhibitors;
Response: Thanks for your comment. We have added information on differences in adverse events between the described PARP inhibitors in lines 318-321, page 10.
- In line 263 please specify where olaparib and talazoparib have been approved (reference for EMA/FDA or other);
Response: Thanks for your comment. We have added the information about where olaparib and talazoparib have been approved in line 312, page 10. Currently, olaparib and talazoparib have been approved by FDA and EMA for the treatment of patients with gBRCAm HER2-negative mBC, and veliparib still in clinical trials.
- In lines 347-352 the reference for immune-related adverse events management is missing. The reference to any guidelines would be better than to KEYNOTE-355 trial;
Response: Thanks for your valuable suggestion. We have carefully checked and replaced it with more appropriate references in line 411, page 13.
Reference
[105] Schneider, B.J.; Naidoo, J.; Santomasso, B.D.; Lacchetti, C.; Adkins, S.; Anadkat, M.; Atkins, M.B.; Brassil, K.J.; Caterino, J.M.; Chau, I.; et al. Management of Immune-Related Adverse Events in Patients Treated With Immune Checkpoint Inhibitor Therapy ASCO Guideline Update. J Clin Oncol 2021, 39, 4073-4126.
Round 2
Reviewer 1 Report
The revised manuscript has made a great improvement. I have no more comments and recommends.